# Depression Recognition Using Daily Wearable-Derived Physiological Data

**DOI:** 10.3390/s25020567

**Published:** 2025-01-19

**Authors:** Xinyu Shui, Hao Xu, Shuping Tan, Dan Zhang

**Affiliations:** 1Department of Psychological and Cognitive Sciences, Tsinghua University, Beijing 100084, China; 2Beijing Huilongguan Hospital, Peking University Huilongguan Clinical Medical School, Beijing 100096, China

**Keywords:** depression, wearable device, multimodal, dynamic, autoregressive

## Abstract

The objective identification of depression using physiological data has emerged as a significant research focus within the field of psychiatry. The advancement of wearable physiological measurement devices has opened new avenues for the identification of individuals with depression in everyday-life contexts. Compared to other objective measurement methods, wearables offer the potential for continuous, unobtrusive monitoring, which can capture subtle physiological changes indicative of depressive states. The present study leverages multimodal wristband devices to collect data from fifty-eight participants clinically diagnosed with depression during their normal daytime activities over six hours. Data collected include pulse wave, skin conductance, and triaxial acceleration. For comparison, we also utilized data from fifty-eight matched healthy controls from a publicly available dataset, collected using the same devices over equivalent durations. Our aim was to identify depressive individuals through the analysis of multimodal physiological measurements derived from wearable devices in daily life scenarios. We extracted static features such as the mean, variance, skewness, and kurtosis of physiological indicators like heart rate, skin conductance, and acceleration, as well as autoregressive coefficients of these signals reflecting the temporal dynamics. Utilizing a Random Forest algorithm, we distinguished depressive and non-depressive individuals with varying classification accuracies on data aggregated over 6 h, 2 h, 30 min, and 5 min segments, as 90.0%, 84.7%, 80.1%, and 76.0%, respectively. Our results demonstrate the feasibility of using daily wearable-derived physiological data for depression recognition. The achieved classification accuracies suggest that this approach could be integrated into clinical settings for the early detection and monitoring of depressive symptoms. Future work will explore the potential of these methods for personalized interventions and real-time monitoring, offering a promising avenue for enhancing mental health care through the integration of wearable technology.

## 1. Introduction

Depression is a common and serious mental disorder leading to continuous effects on patients’ emotional states and daily activities. According to the World Health Organization, over 300 million people are suffering from major depression disorder [1], making it a leading cause of disability and largely aggravating the global disease burden. To handle this worldwide issue, global researchers have introduced technological innovations in the assessment, diagnosis, and management of depression disorders for better mental healthcare. Specifically, providing objective and quantitative information from depression patients was important for improving the accuracy of diagnosis [2,3].

The objective identification of depression using physiological data has emerged as a promising research focus within the field of psychiatry. Researchers have started using brain imaging techniques like fMRI and EEG in resting-state studies and have found the possible neural signatures of depression [4,5]. Task-based studies further extend the paradigm by regarding lab-based tasks, such as video watching, audio listening, etc., as a simplified daily life scenario [6,7,8]. These advancements highlight the significance of exploring the underlying depression-related biomarkers for a more accurate and reliable depression diagnosis. However, the relatively expensive cost and complex settings largely limited the application of current methods. Therefore, a more convenient and low-cost alternative is needed to promote personalized interventions and real-time monitoring in large populations.

Recently, the wide use of wearable devices has opened new possibilities for objective depression diagnosis. About 21% of Americans reported using a smartwatch or fitness tracker in daily life, and the number continues to grow [9]. Usually designed to be low-burden for users, wearable devices can non-invasively measure long-period, real-life data and, therefore, achieve higher ecological validity to monitor mental states compared with lab-based brain imaging techniques [10,11,12]. Using commercial smartwatches, previous research has explored the feasibility of individual trait assessment based on daily heart rate recordings, including personality, mental stress, etc. [13,14]. Furthermore, measurements from wearable devices can provide insights into individuals’ daily states, offering a more comprehensive view of the daily states of depression patients [15]. The latest studies have extended the wearable computing method to clinical use to distinguish patients with depression from those with other diseases [16]. These research studies usually focus on one major physiological indicator such as daily steps [17,18], activity at night (from acceleration) [19], pulse rate variability (from photoplethysmogram, PPG) [20], etc.

Integrating multimodal physiological signals could use the differential information of signals and improve the result of depression assessment [21]. Specifically, multimodal studies bring extra dimensions for insight-enhancing analysis and diversity in proportion to the increase in unique modalities because they are believed to provide related but differential information about mental health [22]. The latest research further highlights the combination of heart rate, skin conductance, and acceleration in depression classification analysis based on daily wearable data [22]. The cardiac responses such as heart rate, as well as skin conductance, are influenced by the autonomic nervous system (ANS) activity stimulated by depression-related emotions, acute stress, etc. [23,24]. The empirical study describes how bodily balance and posture quality degrade with depressive symptoms getting worse and shows the relation between mental health and motion during daily activities [25].

The present study aims to explore the feasibility of depression recognition by extracting multimodal physiological signals derived from wearable devices in daily life scenarios. We hypothesize that individuals’ depression disorder is correlated with their multimodal physiological features. Three major objectives of the present study are summarized as follows: first, to promote the clinical application of objective depression detection, we included clinically diagnosed depressed participants, which was one important step further as compared with previous research mainly with the healthy population [22]; second, to evaluate the feasibility of rapid detection based on low-burden wrist-worn devices, we explored classifications using data with varied durations. Considering the constraints of measurement duration and complexity in settings such as outpatient clinics, this is expected to enhance the potential applicability of the proposed method in clinical scenarios [26,27]. Last, to fully exploit the temporal dynamics of physiological features in depression detection [28], we investigated the effectiveness of dynamic features for classification and compared it to classical static features.

To this end, a daily wearable-derived experiment via a multimodal wristband was conducted to investigate the physiological representation of depression. A custom-designed wristband was used to record physiological data over six-hour daily activities from fifty-eight depression patients. Fifty-eight healthy individuals from a published dataset were selected as healthy controls, with a matched recording length, modality, and gender ratio. To compare the physiological changes between the two groups, we extracted static features such as the mean, variance, skewness, and kurtosis of physiological indicators including heart rate, skin conductance, and acceleration, as well as dynamic features including the autoregressive coefficients reflecting the temporal dynamic of these signals [28]. We adopted representative pattern recognition algorithms frequently used in the previous literature to construct discriminative models for identifying depressed individuals. Furthermore, to demonstrate the application potential of our method in clinical assessment scenarios, we explored shorter data segments by distinguishing depressive individuals on data aggregated over 6 h, 2 h, 30 min, and 5 min segments.

## 2. Materials and Methods

### 2.1. Data Preparation

Our data were composed of 58 depression patients and 58 healthy controls. The patients in our experiment were from the Beijing Huilongguan Hospital’s outpatient department between 2021 and 2022. These patients, aged from 18 to 50 years, were diagnosed by professional psychiatrists at Beijing Huilongguan Hospital based on comprehensive clinical assessments. The physiological data of healthy controls were from a published DAPPER dataset [29], with the same wearable device, group size, and gender ratio matched to the patient group. Specifically, the DAPPER dataset included a five-day recording, and we adopted data from the first day to avoid confounding factors such as device familiarity differing from the patient group.

Each participant was instructed to conduct multimodal physiological measurements during their normal daytime activities from 9:00 to 15:00 for one day by wearing a custom-designed wristband (Psychorus, HuiXin, Beijing, China), as shown in Figure 1a. The wristband, which has been used in several previous studies with daily contexts [29,30,31,32,33], was able to record the signals of acceleration (ACC), skin conductance (SC), and PPG at sampling rates of 20 Hz, 40 Hz, and 20 Hz, respectively, as determined by the device’s built-in hardware. The recorded PPGs were then computed to obtain heart rate (HR) at a 1 Hz sampling rate by an implemented HuiXin software package (version 201708) based on a joint sparse spectrum reconstruction algorithm, which has robust performance against daily activity artifacts [32,33]. Missing data (e.g., unreliable contact) were recognized by the wristband built-in function and marked as a low HR value (HR = 40) [29].

### 2.2. Feature Extraction

Four static features—mean, variance, skewness, and kurtosis—were extracted from each participant to characterize the statistical properties of the physiological indicators, including HR, SC, and ACC across a given data segment. We did not further decompose the SC signal into its tonic and phasic components, mainly for the consideration of the reliability of the decomposition of noisy wearable SC signals in daily contexts [34]. The mean and variance provided insights into the average level and variability of physiological data segments. Skewness quantified the asymmetry of the probability distribution of these features, while kurtosis described the “tailedness” of the distribution or the degree of outliers relative to the central tendency [35]. The inclusion of these higher-order statistical features aligns with previous studies on human emotion recognition and disease diagnosis [36,37]. Prior to the static feature extraction, to ensure relative uniformity among different signal modalities, the SC and ACC signals were down-sampled to match the 1 Hz sampling rate of the HR signal from the wristband [38]. Feature extraction was applied to each data segment, such as the whole 6 h experiment or shorter durations (see Section 2.3). The data segments with more than 50% missing data were excluded in feature extraction and further analysis.

Dynamic features were further extracted to characterize the temporal dynamics of the data segments. To this end, the autoregressive (AR) coefficients of the signals were employed [34]. Specifically, the coefficients were derived from AR models, as depicted in (1), where *X_t_* denotes the signal at time *t*, *a_i_* denotes the coefficient at time lag *i,* and *ε_t_* denotes the random error.(1)$$X_{t}=\sum_{i=1}^{k}a_{i}X_{t-i}+\varepsilon_{t}$$

The order *k* of the AR models represented the number of immediately preceding signals used to predict the present physiological signals. We determined an optimal order by calculating the Akaike Information Criterion (AIC) via “aic” function in MATLAB R2022a, which could minimize predicting errors. Specifically, a lower AIC value represented a better fit for the AR model [39]. The AR models were separately computed for ACC, SC, and HR data, as well as for segments of different durations. The AIC values with model orders from 2 to 8 (corresponding to 2 to 8 s at 1 Hz sampling rate) were calculated separately for each participant and averaged across participants to explore the optimal model while controlling the number of features. The model coefficients *a_i_* from the optimal *k*-order per AR model were extracted as the dynamic features.

### 2.3. Statistical Analysis and Classification

The statistical differences of physiological features between the two groups were first computed. Specifically, independent sample *t*-tests were conducted to assess the gender differences for each physiological dimension. The *p*-values in these multiple comparisons were then re-calculated by the false discovery rate (FDR) method [40], and the corrected *p*-values (*p*_FDR_) were used as the indicator of statistical significance in *t*-tests. Then, we analyzed the statistical differences in physiological features across demographic variables. Independent sample *t*-tests were also conducted between two gender groups (FDR corrected). Additionally, Spearman correlation analyses were performed to assess the relationship between the age of patients and their physiological features (FDR corrected).

Representative classification models were employed to discriminate the participants’ depression results based on these features. Specifically, the following classification models were used according to their promising performance in previous studies focusing on depression recognition, as reported in the existing literature [41], including Random Forest (RF), Support Vector Machine (SVM), K-Nearest Neighbors (KNN), and Linear Discriminant Analysis (LDA). The Random Forest method was employed with the tree number set to 100; the Support Vector Machine method was used with a Gaussian kernel; the K-Nearest Neighbors method was used with the number of nearest neighbors set to 1; the standard Linear Discriminant Analysis was employed with no required hyperparameters. A five-fold cross-participant validation procedure was conducted by circularly splitting the training and testing set, with an equal number of patients and normal people in each set. Each model was trained based on a random 4 out of 5 selection of all participants (92 participants) and tested on the remaining 20% (24 participants) for five-time iterations. The training and test processes were conducted 1000 times. To obtain the final classification accuracies, the predicted accuracies of all iterations were averaged. All the classification models were implemented by using the Statistic and Machine Learning Toolbox in MATLAB R2022a.

To explore the feature contributions in classification models, we adopted different combinations of physiological features in Section 2.2, including (1) all features, including static and dynamic features of three modalities (denoted as ALL); (2) the non-acceleration signal, including static and dynamic features of HR and SC signals (Non-ACC); (3) static features of three modalities (Static); (4) dynamic features of three modalities (Dynamic); (5) dynamic features of non-acceleration signal (Non-ACC Dynamic). To quantificationally explore the statistical significance of the classification results, we compared their classification accuracies with those from random models. The random model was trained with the same feature sets but tested with randomly shuffled depression labels for 1000 times.

To further explore the model stability with shorter data lengths for better clinical applications, we explored the classification models on data aggregated over 6 h, 2 h, 30 min, and 5 min segments, respectively. For the latter three conditions, segments of 2 h, 30 min, and 5 min were randomly selected from the 6 h recording data of each participant in order to avoid potential biases that might be introduced by selecting specific moments. The randomized selections were conducted 1000 times to obtain a distribution of accuracies. We performed feature extraction on all the time segments, including the calculation of both the static and dynamic features. The participants without valid data segments for a specific data duration were excluded from the corresponding classification analysis. For instance, if one participant did not have a valid 2 h data segment after the missing data operation, this participant was excluded from the 2 h classification; however, the same participant may be included for the 5 min classification if he or she had valid 5 min segments.

## 3. Results

In this study, we conducted 6 h recordings from 116 participants, adding up to 696 h recordings (116 participants × 6 h per participant). There were 488.4 h of valid data retained after excluding missing data, with each participant’s valid data proportion ranging from 2.5 (41.7%) to 6.0 (100%) hours. After excluding participants without enough valid data, there were 116, 116, 116, and 112 participants retained in 5 min, 30 min, 2 h, and 6 h conditions in the subsequent individual-level classification analysis.

The across-participant-averaged AIC values for fitting physiological signals with AR models are presented in Figure 2. Based on a comprehensive evaluation of the results from all AR models, order 3 has been empirically selected for feature extraction as it represents a balanced trade-off between the number of features and model performance. Therefore, the following analysis adopts a three-order autoregressive model in the AR feature extractions, and the three-dimensional AR features are denoted as AR1, AR2, and AR3, respectively.

By conducting a Spearman correlation analysis between the age of patients and all physiological features, only one feature (the standard deviation of heart rate) shows a significant correlation (*r* = 0.41, *p* = 0.016) after FDR correction. The independent sample *t*-test of each physiological feature between gender groups is not significant (lowest *p* = 0.39, FDR corrected).

As shown in Table 1, the recording samples from depression patients have significantly lower AR1 (*t*(115) = −2.89, *p* = 0.022, FDR corrected) and higher AR2 (*t*(115) = 2.78, *p* = 0.030) out of HR features than healthy controls. This was identified by conducting independent sample *t*-test, as well as lower kurtosis (*t*(115) = −3.17, *p* = 0.001) and lower AR1 (*t*(115) = −3.69, *p* < 0.001), of SC features. The depression samples also showed significantly lower mean value (*t*(115) = −3.10, *p* = 0.007), higher skewness value (*t*(115) = 2.75, *p* = 0.021), and lower AR coefficients (AR1: *t*(115) = −5.82, *p* = 0.001; AR2: *t*(115) = −9.15, *p* < 0.001; AR3: *t*(115) = −5.99, *p* = 0.001) of the acceleration signal.

Table 2 shows the binary classification accuracies from common pattern recognition models. The RF model finally reports the highest accuracy (90.0 ± 1.7%) with the input of 6 h physiological data, significantly higher than the Random condition (49.7 ± 5.1%). As the input data length decreases from 6 h to 5 min, the classification accuracy declines from 90.0 ± 1.7% to 76.0 ± 3.5%. The other three models also obtain significantly higher results (SVM: 74.1 ± 2.2%; LDA: 81.5 ± 2.3%; KNN: 76.1 ± 2.0%) than corresponding Random conditions.

Table 3 depicts the classification accuracies of the RF model when adopting different input features. The result of the Dynamic feature condition with the 6 h data input is close to the All-feature (ALL) condition, which combines Dynamic and Static features (90.0 ± 1.7% vs. 89.3 ± 1.2%, *p* > 0.05), and is significantly higher than the Static condition (*p* < 0.05). The prediction results of Non-ACC models are lower than ALL condition but significantly higher than Random condition (Non-ACC: 80.5 ± 2.4%; Non-ACC Dynamic: 78.1 ± 2.3%). Figure 3 depicts the distribution of classification accuracies in the ALL-feature condition. While the models with 6 h data achieved the highest accuracy (*p* < 0.05), the models based on 5 min data could also significantly distinguish depressive individuals from healthy controls.

## 4. Discussion

In the present study, we conducted a single-day wearable-derived recording via a multimodal wristband to investigate the physiological representation of depression for objective depression recognition. Our statistical analysis obtained significant correlations between depression and multiple physiological features. By extracting static and dynamic features from three modalities (ACC, SC, and HR), our classification results reported significant accuracy (90.0 ± 1.7%) based on machine learning models in the prediction of whether individuals were depression patients or healthy individuals, demonstrating the feasibility of introducing daily wearable recording to assist clinical depression diagnosis.

The findings that dynamic features obtained superior performance over static features extended the application values of conducting temporal feature extraction in daily recordings. While existing studies have explored the association between mental health and the temporal dynamic of human emotion states [42], our results further demonstrated that the dynamic feature of physiological signals can also make sense in mental health studies, which was not only supported by more significant between-group statistical differences but also by the higher accuracy of dynamic feature-based models in the classification analysis. Additionally, the effect of demographic factors such as age and gender was preliminarily investigated because older adults may exhibit distinct clinical characteristics and biological manifestations of depression from younger individuals [43,44]. However, our statistical analysis did not observe significant variations in physiological features across age and gender variables; therefore, we adopted the whole group into classification models. Nevertheless, considering the limited sample size, future research should conduct quantitative analyses with larger-scale samples to address these potential confounding factors.

The classification results based on distinct combinations of wearable physiological features further indicate that using multimodal measurement data could better reflect the daily representation of depression. Firstly, the subset of physiological modalities was also correlated with depression. While previous research studies have reported that individuals’ daily activity patterns represented by the fluctuation of acceleration were associated with depression symptoms [45,46], the statistical test reports significant differences not only in the acceleration modal but also in the Non-ACC signals (including PPG and SC), corresponding with the fact that Non-ACC models could also obtain significant results in the binary classification. This is possibly due to the less effective control of cognitive function in the depression condition, which reflects the top-down effect from the pre-frontal cortex to the ANS system [47], while it is also possible that depressed states are associated with distinct emotional experience (such as negative emotionality and lower emotional arousal) and corresponding physiological changes in the cardiovascular system [48]. Secondly, as proven by previous research [22], using multimodal data could significantly improve the performance of depression recognition, supported by the highest accuracy obtained in the ALL condition. This result may inspire further exploration of more advanced devices with more physiological modalities in depression recognition studies [49].

Our results across machine learning models and data durations further demonstrated that our approach is promising to be extended to clinical application scenarios. Firstly, while four machine learning models have obtained significant predictive results, the RF model achieves the highest accuracy in our results, which is higher than other EEG-based recognition results [50,51] and peripheral physiological studies in laboratory settings [52]. Considering the widespread use and predictive performance of RF in existing research [53], future studies could explore more advanced forms of RF models to achieve better recognition accuracy, such as applying Weighed Random Forest, Random Forest Artificial Neural Network (RF-ANN), etc. [41,54]. Moreover, while existing studies embracing machine learning algorithm ensembles have reported superior results than single algorithms [55,56], it is preferable to investigate the combination of multiple models in classification analysis. Secondly, the classification accuracy of 76.0% based on a 5 min short-duration measurement showed great clinical potential for fast and reliable depression diagnosis, significantly reducing the burden on both patients and healthcare providers. This finding highlights the feasibility of using short recordings for objective assessments. One possible explanation is the long-term influence of a depressed mood, which may create lasting physiological signatures detectable within a brief measurement. Another explanation is that individuals with depression exhibit distinct activity patterns compared to healthy controls. Future research should delve into these underlying factors to further advance wearable technologies for depression recognition. This result could be due to the long-term influence of the depressed mood; another explanation is that individuals diagnosed with depression share distinct activity patterns from normal healthy people. Further studies should explore the underlying factors of the significant result from short-duration recordings to better promote wearable depression recognition.

There are some limitations that should be noted for further studies besides the sample size and machine learning models. Firstly, the generalizability of this method may be limited by one-day measurement. Since individuals’ depression states could fluctuate in daily life [57], it is more appropriate to collect longitudinal wearable data to investigate the temporal association between mental health and physiological indicators that contributes to daily mental health monitoring. Secondly, individuals’ physiological signals have been known to vary substantially across time [58], which might influence the accuracy of classification models. While our findings have demonstrated the robustness of the proposed method for the classification of randomly selected time segments, it is necessary to conduct a multi-day recording to further evaluate the long-term stability of the proposed method across days, weeks, or longer. Thirdly, the feature extraction strategy in this study was relatively conservative, focusing on classical features from these physiological signals. A further exploration of feature extraction could be beneficial for depression recognition, including the analysis of the raw PPG signals [59], the decomposition of skin conductance components [60], and more advanced neural-network-based analyses [61,62]. Another limitation is that the result was restricted to the binary classification of depression or health. Considering the current experience of digital phenotype in depression studies [63], this method could be similarly extended further and output more detailed information, such as the degree of depression.

In sum, our results demonstrate the feasibility of using wearable devices for depression recognition through a lightweight wristband that is representative of the capability of state-of-the-art lightweight devices. The continuous recording of physiological data throughout the daily events allows for a more comprehensive understanding of individual variations in depression-specific representations [57], which could further contribute to more suitable mental health care through the integration of wearable technology, such as personalized intervention, real-time monitoring, etc.

## Figures and Tables

**Figure 1 sensors-25-00567-f001:**
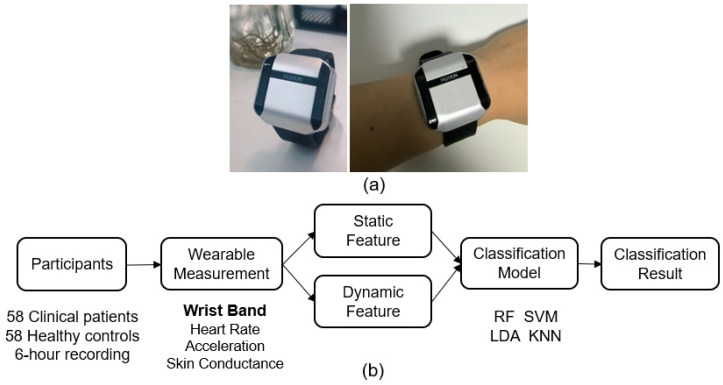
(**a**) The wristband device; (**b**) the flowchart of data analysis. The classification models include Random Forest (RF), Support Vector Machine (SVM), K-Nearest Neighbors (KNN), and Linear Discriminant Analysis (LDA).

**Figure 2 sensors-25-00567-f002:**
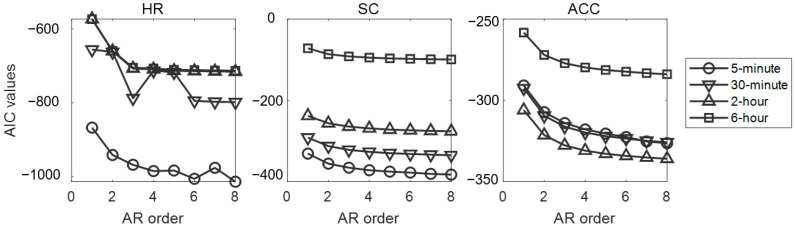
The overall AIC criterion of all participants across AR models with different indexes. Three subplots, respectively, represented the AIC in AR models based on HR, SC, and ACC signals.

**Figure 3 sensors-25-00567-f003:**
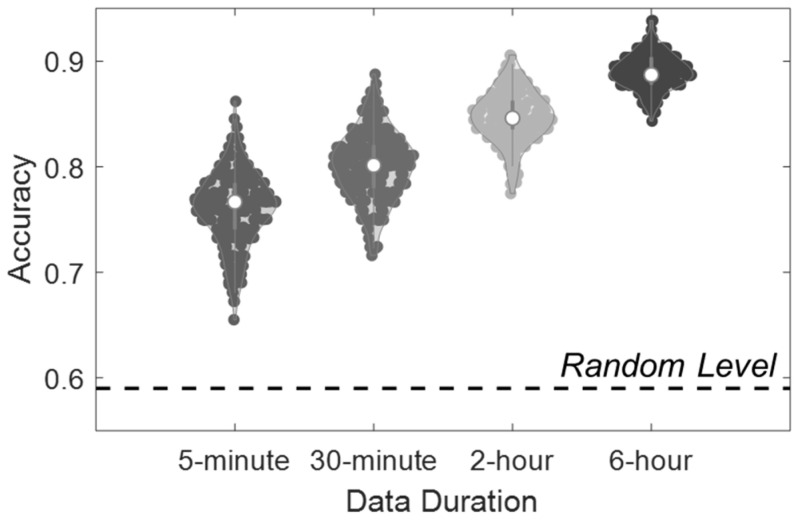
The violin plot shows the distribution of classification accuracies based on bootstrapped RF models in the ALL-feature condition. In each bootstrap process, data segments were randomly selected and split (of training and test sets) in the 5 min, 30 min, 2 h, and 6 h conditions. Each dot showed the averaged 5-fold accuracy from one out of 1000 bootstrap steps. The dotted line represented the 95% interval of the Random condition results.

**Table 1 sensors-25-00567-t001:** Independent *t*-test results between depression patients and healthy controls.

Features		Heart Rate		Skin Conductance		Acceleration	
		*t*(115)	*p_FDR_*	*t*(115)	*p_FDR_*	*t*(115)	*p_FDR_*
Static	Mean	−0.05	0.963	−0.59	0.673	−3.10	0.007
	SD	−1.43	0.298	−1.11	0.408	0.14	0.932
	Skew	−0.68	0.655	0.56	0.673	2.75	0.021
	Kurt	−0.68	0.655	−3.17	0.001	−1.10	0.408
Dynamic	AR1	−2.89	0.022	−3.69	<0.001	−5.82	0.001
	AR2	2.78	0.030	−1.24	0.383	−9.15	<0.001
	AR3	−2.09	0.082	−0.16	0.932	−5.99	0.001

The *p*-value has been corrected by the FDR method.

**Table 2 sensors-25-00567-t002:** Classification accuracy based on different models across distinct data segments.

Models	5 min	30 min	2 h	6 h	Random
RF	76.0 (3.5)	80.1 (3.2)	84.7 (2.5)	90.0 (1.7)	49.7 (5.1)
SVM	67.4 (4.4)	67.0 (4.0)	73.2 (3.5)	74.1 (2.2)	49.2 (4.5)
LDA	64.6 (5.1)	69.2 (4.7)	77.0 (3.4)	81.5 (2.3)	52.1 (3.7)
KNN	60.8 (4.9)	64.6 (5.5)	71.6 (3.7)	76.1 (2.0)	53.4 (2.6)

Each column showed the classification accuracy denoted by mean (SD) % of models based on specific data lengths, such as 5 min, 30 min, 2 h, and 6 h physiological data segments. The “Random” column showed a random accuracy level obtained by randomly shuffling participants’ depression labels based on corresponding 6 h models. The classification models include Random Forest (RF), Support Vector Machine (SVM), K-Nearest Neighbors (KNN), and Linear Discriminant Analysis (LDA).

**Table 3 sensors-25-00567-t003:** Classification accuracies based on Random Forest models across distinct feature sets.

Selected Features	5-min	30-min	2-h	6-h	Random
ALL	76.0 (3.5)	80.1 (3.2)	84.7 (2.5)	90.0 (1.7)	49.7 (5.1)
Non-ACC	70.4 (4.8)	74.6 (4.4)	77.0 (3.8)	80.5 (2.4)	50.4 (2.0)
Static	75.8 (3.2)	78.1 (3.1)	80.6 (3.3)	85.7 (1.5)	49.9 (2.4)
Dynamic	74.5 (3.7)	77.3 (3.5)	82.5 (2.9)	89.3 (1.2)	51.2 (2.3)
Non-ACC Dynamic	68.4 (4.6)	71.5 (4.3)	74.5 (3.8)	78.1 (2.3)	52.3 (4.0)

Each column showed the classification accuracy denoted by mean (SD) % of models based on specific data lengths, such as 5 min, 30 min, 2 h, and 6 h physiological data segments. “Random” column showed a random level of accuracy by shuffling participants’ depression labels. The selected features from row 1 to row 5 represent (1) all features (ALL); (2) the non-acceleration signal, including HR and SC signals (Non-ACC); (3) static features (Static); (4) dynamic features (Dynamic); (5) dynamic features of non-acceleration signals (Non-ACC Dynamic).

## Data Availability

The raw data supporting the conclusions of this article will be made available by the authors on request.

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
