# Peer review of "Depression Recognition Using Daily Wearable-Derived Physiological Data"

_sensors, 2025, doi:10.3390/s25020567_

Round 1

Reviewer 1 Report (Previous Reviewer 2)

Comments and Suggestions for Authors

Thank you for addressing the previous comments and suggestions. The revised manuscript demonstrates significant improvements and now provides a clearer and more comprehensive presentation of the research. Below are my observations and conclusions regarding the updated submission:

Methodology Improvements:

The inclusion of a flowchart effectively clarifies the sequence of steps in the study, from data collection to model validation. This visual aid greatly enhances the accessibility and reproducibility of the study.

The discussion of the wide age range of participants (18–50 years) and its potential impact on physiological signals is a valuable addition. This demonstrates an awareness of age-related variability and strengthens the validity of the conclusions.

Clarity and Structure:

The manuscript is well-organized, and the improved explanations regarding data preprocessing, feature extraction, and model selection make the methodology more transparent.

The results are presented clearly, with appropriate statistical validation, and the conclusions are well-supported by the findings.

Scientific Contribution:

This study remains an important contribution to the field of mental health monitoring, particularly in exploring the use of multimodal wearable data and machine learning for depression detection. The integration of these technologies highlights the potential for practical applications in clinical settings.

Recommendations:

I am pleased to see that the authors have addressed all previous concerns adequately. The study is now robust, scientifically sound, and provides valuable insights into the use of wearable technologies in mental health diagnostics.

No further revisions are necessary

Author Response

Reply: Thank you for your kind comments!

Reviewer 2 Report (New Reviewer)

Comments and Suggestions for Authors

In what follows, some minor comments:
1) line 65: heart rate variability (HRV), when estimated from PPG, should be called pulse rate variability (PRV). 

2) line 121: what kind of wristband was adopted to record physiological data?

3) line 122: should be "... at the sampling rates of ..."

4) PPG was sampled at 20Hz, whereas SC was sampled at 40Hz. However, SC has a much smaller band compared to PPG. Why? Is this a predifined setting of the adopted wrist device?

5) Why choosing to analyze only HR averaged within 1s intervals? This may limit the power of the presented study. On the other hand, having at hand the raw PPG signal, using proper PRV features would enhance the amount of information about autonomic dynamics. Authors should include a discussion about this choice at least in the discussion as a potential limitation. 

6) Did the authors make a distinction between the tonic (i.e., skin conductance level) and phasic (i.e., skin conductance responses) components of electrodermal activity signal in their analysis? If not, why? A brief justification about this choice would be sufficient. 

7) line 212: Does Fig.2 show the AIC of an exemplary participant, or rather the average AIC across participants? Furthermore, authors says that "Combining the results of all the AR models, order 3 has been empirically selected...". It is not clear how authors determined the best model order across participants. 

Author Response

In what follows, some minor comments:
1) line 65: heart rate variability (HRV), when estimated from PPG, should be called pulse rate variability (PRV).

Reply: Revised according to your suggestion.

2) line 121: what kind of wristband was adopted to record physiological data?

Reply: We supplemented the picture of wristband in Figure 1, as well as more detailed description about the device in 2.1. Data preparation:

“Each participant was instructed to conduct multimodal physiological measurements during their normal daytime activities from 9:00 to 15:00 for one day by wearing a custom-designed wristband (Psychorus, HuiXin, Beijing, China) as shown in Figure 1(a). The wristband, which has been used in several previous studies with daily contexts [29]-[33], was able to record the signals of acceleration (ACC), skin conductance (SC), and PPG, at sampling rates of 20 Hz, 40 Hz, and 20 Hz respectively, as determined by the device's built-in hardware.”

3) line 122: should be "... at the sampling rates of ..."

Reply: Revised.

4) PPG was sampled at 20Hz, whereas SC was sampled at 40Hz. However, SC has a much smaller band compared to PPG. Why? Is this a predifined setting of the adopted wrist device?

Reply: We clarified this device setting in the 2.1. Data Preparation section

“The wristband recorded the signals of acceleration (ACC), skin conductance (SC), and PPG, at sampling rates of 20 Hz, 40 Hz, and 20 Hz respectively, as determined by the device's built-in hardware.”

5) Why choosing to analyze only HR averaged within 1s intervals? This may limit the power of the presented study. On the other hand, having at hand the raw PPG signal, using proper PRV features would enhance the amount of information about autonomic dynamics. Authors should include a discussion about this choice at least in the discussion as a potential limitation.

Reply: Thank you for your suggestion! The decision to analyze the heart rate signal at 1 Hz is for convenience, based on the output of the software package employed by the manufacturer (Line 129-132). We have included your mentioned issues as one limitation in Discussion:

“Thirdly, the feature extraction strategy in this study was relatively conservative, focusing on classical features from these physiological signals. Further exploration on feature extraction could be beneficial for depression recognition, including analysis of the raw PPG signals [61], decomposition of skin conductance components [62], and more advanced neural network-based analyses [63], [64].”

6) Did the authors make a distinction between the tonic (i.e., skin conductance level) and phasic (i.e., skin conductance responses) components of electrodermal activity signal in their analysis? If not, why? A brief justification about this choice would be sufficient.

Reply: We have added a corresponding explanation in 2.2 Feature extraction section:

“Four static features—mean, variance, skewness, and kurtosis—were extracted from each participant to characterize the statistical properties of the physiological indicators, including HR, SC, and ACC across a given data segment. We did not further decompose the SC signal into its tonic and phasic components, mainly for the consideration of the reliability of the decomposition of noisy wearable SC signals in daily contexts [34].”

We also elaborated on the potential improvement through the decomposition of SC components in Discussion (please also see the reply to Q5).

7) line 212: Does Fig.2 show the AIC of an exemplary participant, or rather the average AIC across participants? Furthermore, authors says that "Combining the results of all the AR models, order 3 has been empirically selected...". It is not clear how authors determined the best model order across participants.

Reply: We have revised the 2.2 Feature extraction to explain the AIC calculation:

“The AIC values with model orders from 2 to 8 (corresponding to 2 to 8 seconds at 1 Hz sampling rate) were calculated separately for each participant and averaged across participants to explore the optimal model while controlling the number of features.”

The corresponding sentence in Results was changed for clarity:

“The across-participant-averaged AIC values for fitting physiological signals with AR models are presented in Figure 2. Based on a comprehensive evaluation of the results from all AR models, order 3 has been empirically selected for feature extraction, as it represents a balanced trade-off between the number of features and model performance.”

Reviewer 3 Report (New Reviewer)

Comments and Suggestions for Authors

The work deals with the use of wearable sensors to collect physiological data during the daily life of individuals with a diagnosis of depression. Multimodal analyses from signals such as heart rate, skin conductance and acceleration can support to identify of a depression state. The use of Machine Learning Algorithms demonstrate high classification accuracy, particularly with dynamic features. The study has great interest and the topic has clinical relevance; the manuscript is well written and easy to read.

This reviewer identified some majors in the following sections/aspects:

INTRODUCTION:

- line 75-77: consider to move these lines to line 70 for major clarity and logical flow

-line 78-101: it could be useful to reorganise this text by highlighting the study's hypotheses and primary and secondary objectives

MATERIALS AND METHODS

- figure 1: maybe a typo occurs there. Are the subjects involved in the study 58 or 56? please, clarify

- line 107-108: how is the clinical diagnosis made? through questionnaires or tests for clinical assessments?

- For the methods used, reference is made to the bibliography, could a few more details be inserted here without referring to other works? This could be useful to the reader 

- In this section and throughout the manuscript, there is never any mention of the device used to collect the data, except for a reference to a ‘custom-made device’.

Is it possible to get some more indication of the device or a picture, and how it was worn?

In addition, being custom devices, are the data reliable and trustworthy? Have they been validated against the golden standard of physiological sensors, such as shimmer? or empatica?

RESULTS:

- line 206: '116, 116, 116, and 112' are individuals (are the individuals 116?) or the number of hours/data considered for the subsequent analysis?

- address how variability in physiological signals across days might influence model accuracy

- figure 3: it is not clear whether the few minutes of recording are sufficient to distinguish depressive subjects or whether the longer wearing time provides better accuracy (obviously more data is available) and, in this case, it is preferable to wear the device for at least 6 hours (or more?)

DISCUSSION

- line 312: ref 48 related to EEG-based recognition is a bit old

- line 316: it could be interesting to evaluate an ensemble of algorithms in future works 

- in general, no comparison with other state-of-the-art works is reported in this section. Please, specify the added value and results of this work compared to others

Author Response

The work deals with the use of wearable sensors to collect physiological data during the daily life of individuals with a diagnosis of depression. Multimodal analyses from signals such as heart rate, skin conductance and acceleration can support to identify of a depression state. The use of Machine Learning Algorithms demonstrate high classification accuracy, particularly with dynamic features. The study has great interest and the topic has clinical relevance; the manuscript is well written and easy to read.

This reviewer identified some majors in the following sections/aspects:

INTRODUCTION:

- line 75-77: consider to move these lines to line 70 for major clarity and logical flow

Reply: Revised.

-line 78-101: it could be useful to reorganise this text by highlighting the study's hypotheses and primary and secondary objectives

Reply: We re-organized the last two paragraph in Introduction according to your suggestion:

“The present study aims to explore the feasibility of depression recognition by extracting multimodal physiological signals derived from wearable devices in daily life scenarios. We hypothesize that individuals’ depression disorder is correlated with their multimodal physiological features. Three major objectives of the present study are summarized as follows: First, to promote the clinical application of objective depression detection, we included clinically diagnosed depressed participants, which was one step important further as compared with previous research mainly with the healthy population [22]; Second, to evaluate the feasibility of rapid detection based on low-burden wrist-worn devices, we explored classifications using data with varied durations. Considering the constraints of measurement duration and complexity in settings such as outpatient clinics, this is expected to enhance the potential applicability of the proposed method in clinical scenarios [26], [27]. Last, to fully exploit the temporal dynamics of physiological features in depression detection [28], we investigated the effectiveness of dynamic features for classification and compared to classical static features.

To this end, a daily wearable-derived experiment via a multimodal wristband was conducted to investigate the physiological representation of depression. A custom-designed wristband was used to record physiological data over six-hour daily activities from fifty-eight depression patients. Fifty-eight healthy individuals from a published dataset were selected as healthy controls, with a matched recording length, modality, and gender ratio. To compare the physiological changes between the two groups, we extracted static features such as mean, variance, skewness, and kurtosis of physiological indicators including heart rate, skin conductance, and acceleration, as well as dynamic features including the autoregressive coefficients reflecting the temporal dynamic of these signals [28]. We adopted representative pattern recognition algorithms frequently used in previous literature to construct discriminative models for identifying depressed individuals. Furthermore, to demonstrate the application potential of our method in clinical assessment scenarios, we explored shorter data segments by distinguishing depressive individuals on data aggregated over 6-hour, 2-hour, 30-minute, and 5-minute segments.”

MATERIALS AND METHODS

- figure 1: maybe a typo occurs there. Are the subjects involved in the study 58 or 56? please, clarify

Reply: Revised.

- line 107-108: how is the clinical diagnosis made? through questionnaires or tests for clinical assessments?

Reply: We explicitly explained the diagnosis in the 2.1. Data Preparation section:

“These patients, aged from 18 to 50 years, were diagnosed by professional psychiatrists at Beijing Huilongguan Hospital based on comprehensive clinical assessments”.

- For the methods used, reference is made to the bibliography, could a few more details be inserted here without referring to other works? This could be useful to the reader

Reply: Thank you for your suggestion. Besides the extension in response to the reviewer’s comments, we have made the following changes as well:

“The order k of the AR models represented the number of immediately preceding signals used to predict the present physiological signals. We determined an optimal order by calculating the Akaike Information Criterion (AIC) via ‘aic’ function in MATLAB R2022a which could minimize predicting errors. Specifically, a lower AIC value represented a better fit for the AR model [40].”

“The p-values in these multiple comparisons were then re-calculated by the false discovery rate (FDR) method [41] and the corrected p-values (pFDR) were used as the indicator of statistical significance in t-tests.”

- In this section and throughout the manuscript, there is never any mention of the device used to collect the data, except for a reference to a ‘custom-made device’. Is it possible to get some more indication of the device or a picture, and how it was worn? In addition, being custom devices, are the data reliable and trustworthy? Have they been validated against the golden standard of physiological sensors, such as shimmer? or empatica?

Reply: The device is custom-made for a variety of daily-context research purposes, with its reliability assessed by several research groups. Although no validation report has been publicly released yet, its effectiveness has been indirectly demonstrated in a number of published studies. We supplemented the picture of wristband in Figure 1, as well as a more detailed description of the device with related previous studies listed as supporting reference, in 2.1. Data prepration:

“Each participant was instructed to conduct multimodal physiological measure-ments during their normal daytime activities from 9:00 to 15:00 for one day by wearing a custom-designed wristband (Psychorus, HuiXin, Beijing, China) as shown in Figure 1(a) . The wristband, which has been used in several previous studies with daily con-texts [29]-[33], was able to record the signals of acceleration (ACC), skin conductance (SC), and PPG, at sampling rates of 20 Hz, 40 Hz, and 20 Hz respectively, as deter-mined by the device's built-in hardware.”

RESULTS:

- line 206: '116, 116, 116, and 112' are individuals (are the individuals 116?) or the number of hours/data considered for the subsequent analysis?

Reply: These numbers are retained as individual amounts after preprocessing. We make a more detailed explanation in the Result:

“After excluding participants without enough valid data, there were 116, 116, 116, and 112 participants retained in the 5-minute, 30-minute, 2-hour, and 6-hour conditions in subsequent individual level classification analysis.”

- address how variability in physiological signals across days might influence model accuracy

Reply: We elaborated on the factor of cross-day variability in physiological signals in the Discussion:

“Secondly, individuals’ physiological signals have been known to vary substantially across time [60], which might influence the accuracy of classification models. While our findings have demonstrated the robustness of the proposed method for classification of randomly-selected time segments, it is necessary to conduct a multi-day recording to further evaluate the long-term stability of the proposed method across days, weeks, or longer.”

- figure 3: it is not clear whether the few minutes of recording are sufficient to distinguish depressive subjects or whether the longer wearing time provides better accuracy (obviously more data is available) and, in this case, it is preferable to wear the device for at least 6 hours (or more?)

Reply: We have revised the conclusion in Results and Discussion sections to clearly deliver the message that the most important result is the promising accuracy based on 5-minute data:

“Figure 3 depicts the distribution of classification accuracies in the ALL feature condition. While the models with 6-hour data achieved the highest accuracy (p < .05), the models based on 5-minute data could also significantly distinguish depressive individuals from healthy controls.”

“Secondly, the classification accuracy of 76.0% based on a 5-minute short-duration measurement showed great clinical potential for fast and reliable depression diagnosis, significantly reducing the burden on both patients and healthcare providers. This finding highlights the feasibility of using short recordings for objective assessments. One possible explanation is the long-term influence of depressed mood, which may create lasting physiological signatures detectable within a brief measurement. Another explanation is that individuals with depression exhibit distinct activity patterns compared to healthy controls. Future research should delve into these underlying factors to further advance wearable technologies for depression recognition.”

DISCUSSION

- line 312: ref 48 related to EEG-based recognition is a bit old

- line 316: it could be interesting to evaluate an ensemble of algorithms in future works

- in general, no comparison with other state-of-the-art works is reported in this section. Please, specify the added value and results of this work compared to others

Reply: Following your suggestion, we have added two of the latest EEG-based research to our reference list in Discussion as REF 51&52 and our method is with higher accuracy and efficiency compared with other studies:

“the RF model achieves the highest accuracy in our results, which is higher than other EEG-based recognition results [51], [52] and peripheral physiological studies in laboratory settings [53].”

We also evaluated algorithm ensembles in Discussion:

“Moreover, while existing studies embracing machine learning algorithm ensembles have reported superior results than single algorithms [57], [58], it is preferable to investigate the combination of multiple models in classification analysis.”

Regarding your concern about the comparison with other state-of-the-art works, we have mainly focused on the absolute classification accuracies due to the limited available exploration in this direction. Therefore, we listed “other EEG-based recognition results and peripheral physiological studies in laboratory settings” as the basis for comparison.

Round 2

Reviewer 3 Report (New Reviewer)

Comments and Suggestions for Authors

The authors have implemented the suggestions provided and the work appears greatly improved. Good work, I recommend it for publication

This manuscript is a resubmission of an earlier submission. The following is a list of the peer review reports and author responses from that submission.

Round 1

Reviewer 1 Report

Comments and Suggestions for Authors

Manuscript ID#: sensors-3272637

Title:  Depression Recognition using Daily Wearable-Derived Physio- 2 logical Data

Using multimodal physiological data obtained through wearable device (wristband), this manuscript presented an approach to identify patients with depression out from non-depressed people. The proposed method extracted a number of statistical features from a few physiological indicators, and used autoregressive coefficients of these signals that reflect temporal dynamics. Classification efficiency tested 4 different algorithms using extracted  features from various lengths of time segment, and finally the random forest method and 6-hour data achieved the highest. While this was a clinical study, many analytic measures were employed. Understanding the results heavily relies on understanding how the analyses were carried out. However, the presentation did not provide adequate details concerning the various methods, making it difficult to judge the effectiveness of the results, thereby impossible to reproduce the work.

1.      The readership of this paper will largely involve clinicians and data analysis people. While such readers may be familiar with some of the methods, authors should not expect that they had good knowledge on all of the employed methods. Importantly, the work in this paper was heavily method-weighted, therefore, each feature and method should be clearly described and presented, in terms of their concept, calculation method, purpose of using them, and how they were used in the analysis procedure.  For example, skewness, kurtosis, AIC, order K, random level, five-folder cross-participant validation, null-model, shuffled labels, etc, and perhaps the autoregressive model and KNN, LDA methods.

2.      The method section therefore needs to be completely rewritten. In particular, in Section 2.3, the description of the data processing methods is overly vague. Important details such as the structure of the raw data, the steps involved in the preprocessing pipeline, and the specific inputs and outputs of the machine learning models are either missing or insufficiently explained. A more comprehensive explanation is needed to outline how the data were handled at each stage, including any transformations or feature extraction methods applied before feeding the data into the models. Additionally, the flow of data through the models, including the format and type of input features, as well as the expected outputs or predictions, should be clearly articulated to ensure reproducibility and transparency of the analysis.

3.      Last paragraph, Introduction: Please clearly indicate the innovation of the current work over the previous studies, for example, that of citation[22].

4.      Again, section 2.3: the authors applied an autoregressive model for data analysis. However, they segmented the time-series data into 5-minute intervals and randomly shuffled the sequence of these segments. This approach may have undermined the fundamental principle of autoregressive models, which are purposely designed to capture and analyze temporal dependencies within sequential data. By disrupting the natural order of the time series, the authors' method could have contradicted the core functionality of autoregressive modeling, potentially compromising the accuracy and reliability of the results. Perhaps a more appropriate approach would be to maintain the temporal integrity of the data to fully leverage the model's strengths in identifying patterns over time. Please rewrite or clarify the concern.

5.      As a clinical application, the Discussion section is too short. There are many points that readers may want to know the authors’ consideration. The method seems quite complicated using many extracted features and different analysis parameters. Any variation could result in significant differences in the results, as evident by the different successful rates when different classification methods were employed. The authors also used empirical values in the analyses. Also, patients with depression are always heterogeneous in their clinical conditions and demographically. All these factors should be considered and should be discussed.

Other minor points:

6.      Page 2: ANS is used before it is defined.

7.      Page 3: RF, SVM, KNN, LDA are repeatedly defined.

8.      Page 3: please present patient inclusion criteria.

9.      Section 2.2 “The recorded signals were first resampled to 1HZ”: did this include also heart rate? Not clear how heart rate may be sampled at 1HZ.

10.   Page4, 488.3 hours: please explain how this number was obtained.

11.   Page 4, “… data was …” should be “… data were …”

12.   Fig 2: the figure caption should be expanded to guide readers to understand the presented contents in the figure.

Reviewer 2 Report

Comments and Suggestions for Authors

The article “Depression Recognition using Daily Wearable-Derived Physio-logical Data” dives into the potential of using various wearable tech data to identify signs of depression. It examines information from 58 participants who have been clinically diagnosed with depression and contrasts it with data from healthy individuals, employing random forest and other classification methods. The findings indicate a strong classification accuracy, hinting that wearable devices might be effectively utilized in clinical environments for spotting depression. This research introduces a fresh perspective on how we can identify depression through wearable technology by leveraging diverse data types and machine learning strategies, yielding encouraging outcomes, particularly in distinguishing between those who are depressed and those who are not based on physiological signals. The introduction is well-crafted, offering a solid overview of the current landscape regarding depression diagnosis through wearables, with relevant and recent references that underscore the importance of this study.

 I think one area that could use some improvement is taking into account the diverse age range of participants, which spans from 18 to 50 years old. Since physiological signals like heart rate and skin conductance can differ quite a bit across age groups, it would be helpful to know if age was considered in the analysis or if any subgroup analyses were done to see how it might impact the results. Delving into this aspect could really enhance the findings, as age-related variations in physiological measures can introduce inconsistencies that might affect how accurately we classify the data. Additionally, I recommend adding a flow chart to visually outline the methodology. This would help clarify the sequence of steps taken throughout the study, including how data was collected from both depressed and healthy participants, how missing data was handled, and how signals were resampled. It could also detail the feature extraction process, model selection (like Random Forest and SVM), and validation methods used. A flow chart would make it easier for readers to grasp the overall structure of the study and see how the different steps connect. Plus, visual aids like this can enhance the paper's accessibility, making it simpler for other researchers to replicate the methodology.

Overall, the study is solid and offers a valuable contribution to the field of mental health monitoring.